# Vaccine Uptake and COVID-19 Frequency in Pregnant Syrian Immigrant Women

**DOI:** 10.3390/vaccines11020257

**Published:** 2023-01-25

**Authors:** Mehmet Akif Sezerol, Zeynep Meva Altaş

**Affiliations:** 1Doctorate Program, Institute of Health Sciences Epidemiology, Medipol University, Istanbul 34810, Turkey; 2Health Management Program, Graduate Education Institute, Maltepe University, Istanbul 34857, Turkey; 3Sultanbeyli District Health Directorate, Istanbul 34935, Turkey; 4Ümraniye District Health Directorate, Istanbul 34764, Turkey

**Keywords:** immigrants, vaccination, pregnant women, COVID-19

## Abstract

Immigrants have difficulties in the use of essential health services such as vaccinations. Vaccine uptake among pregnant immigrant women is very low. The aim of the study was to examine the vaccination status of pregnant immigrant women who received health services in an immigrant health center (IHC) affiliated to primary health care institutions. The research is a retrospective-designed cross-sectional type of study. The study sample consists of pregnant Syrian women who received health care from the strengthened IHC of a District Health Directorate in Istanbul between August 2020 and 2022. Age, trimesters, number of pregnancies, high-risk pregnancy status, vaccination dates and status against influenza, COVID-19 and tetanus, and vaccine types of COVID-19 were evaluated. The statistical significance level was determined as *p* < 0.05. None of the pregnant women had received the influenza vaccine. Of the women whose tetanus vaccine data were evaluated, 29.7% had received at least two doses of the tetanus vaccine. Of the pregnant women, 19.4% were vaccinated against COVID-19 with a minimum two doses and 4.2% had a COVID-19 infection during their pregnancy. None of the women with the COVID-19 infection were fully vaccinated against COVID-19. The vaccine uptake of pregnant immigrant women is very low. Public health interventions are needed to improve vaccination coverage among disadvantaged groups.

## 1. Introduction

Vaccines prevent many life-threatening diseases and provide a longer and healthier life. According to the World Health Organization, 3.5–5 million deaths every year are prevented with immunization [1]. Vaccination recommendations vary according to different populations such as children, adults and pregnant women [2].

Since pregnant women are among the high-risk groups in terms of some infections such as influenza, vaccination of pregnant women is extremely important for both maternal and fetal health [3]. According to the recommendation of Centers for Disease Control and Prevention (CDC) [4], women should be vaccinated with inactivated influenza and Tdap (Tetanus–Diphtheria–Acellular Pertussis) vaccines in each pregnancy [5]. In Turkey, at least two doses of the tetanus toxoid vaccine Td (Tetanus–Diphtheria) is recommended by the Ministry of Health after 12 weeks of gestation for all pregnant women who are unvaccinated against tetanus [6]. Additionally, one dose of a seasonal influenza vaccine at any time of pregnancy is also recommended for pregnant women [7]. Being infected with COVID-19 during pregnancy causes an increased risk of complications that can affect both pregnant women and the developing baby [8]. The American College of Obstetricians and Gynecologists (ACOG) has recommended COVID-19 vaccines to pregnant women since early 2021 [9]. The Royal College of Obstetricians and Gynaecologists (RCOG) states pregnant women receiving two doses and the booster dose are less likely to be hospitalized due to COVID-19 than those who are unvaccinated [10].

Four authorized COVID-19 vaccines are available worldwide: BNT162 (Pfizer BioNTech, New York, NY, USA), ChAdOx1 (AstraZeneca, Oxford, UK), mRNA1273 (Moderna, Cambridge, MA, USA) and Ad26.COV2-S (Johnson & Johnson, New Brunswick, NJ, USA). In addition, there are other vaccines, such as BBIBP-CorV (Sinopharm, Beijing, China), CoronaVac (Sinovac, Beijing, China), Sputnik V (Gamaleya, Moscow, Russian) and COVAXIN (Bharat Biotech, Hydrabad, India), which are also authorized for use in many countries [11]. The immune system begins to react after vaccination, and side effects may be seen due to the response of immune system. The risk of possible side effects are higher in mRNA vaccines [12]. According to a systematic review of the literature, COVID-19 vaccination in pregnant and lactating mothers was found to be effective in preventing COVID-19 infection, and it was stated that the vaccine did not cause significant side effects or obstetric, neonatal outcomes [13]. In another study, post-vaccination side effects were similar in pregnant and non-pregnant women who received the COVID-19 vaccine. No serious post-vaccine reactions were observed in either pregnant and non-pregnant women. [14]. A systematic review reported that local pain, fever, chills, fatigue, myalgia, arthralgia, headache, nausea, decreased sleep quality and palpitations were some of the side effects after COVID-19 vaccination among pregnant women [15]. Side effects occurring after vaccination can cause an increase in vaccine hesitancy in many people [16]. Increasing the knowledge about vaccine efficacy and the true side effects are necessary to enhance vaccine uptake [16]. 

COVID-19 vaccination activities have started gradually in Turkey as in the rest of the world. Firstly, healthcare workers, the elderly and those with chronic diseases were vaccinated, and then vaccination continued in all age groups. Since the end of the June 2021, all adult age groups have been granted free vaccination by the Ministry of Health. CoronaVac (Sinovac), Pfizer-BioNTech and Turkovac vaccines are currently in use in our country and these are applied free of charge for all, including immigrants.

Every pregnant woman should have the right for special care and assistance during pregnancy, delivery, and the postpartum period [17]. However, difficulties and inequality in the use of health services by immigrants can be seen due to language problems, economic inadequacy, low education level, etc. [18,19]. Vaccination is one of the essential health services that cannot be used to an adequate level by immigrant women [20]. However, vaccination is an important and an indisputable human right within primary health care services [1]. 

Differences in vaccine acceptance and attitudes towards vaccination can lead to inequalities in the percentages of vaccine uptake [21]. Evidence from the literature suggests that vaccine uptake among pregnant immigrant women is very low. According to a systematic review and meta-analyses, 27.5% (95% CI: 18.8–37.0%) of pregnant women were vaccinated against COVID-19 [22]. According to the results of the same study, the percentage of COVID-19 vaccination was reported to be lower in ethnic minority groups [22]. According to a study conducted in France, 7.4% of pregnant women received the influenza vaccine. In the same study, this percentage was found to be lower in pregnant women born outside of France [23]. The survey on the health status of the Syrian refugee population in Turkey conducted by the World Health Organization indicates that the percentage of pregnant women who are not vaccinated against tetanus is more than four times that of vaccinated pregnant women [24].

As Turkey is a European country but is not a member of either the European Union or the European Free Trade area, Turkey has often been omitted from research on COVID-19 vaccine coverage on the continent. However, Turkey deserves to be a big focus of this topic. A large number of immigrants have come to Turkey in recent years [25]. Syrian immigrants are also among these groups. The size of the Syrian immigrant population in our country has been increasing since 2011. The number of Syrian women in Turkey is 1,655,111, as of December 2022, and almost half of them (796.429) are of childbearing age (15–49 years) [26]. For this reason, it is extremely important to evaluate the health status of pregnant immigrant women and the extent of the benefit from health services that they need to access during pregnancy, such as vaccination. In this way, health interventions can be planned for the required areas. In this study, the aim was to examine the recommended vaccination status of pregnant immigrant women who received health services in an immigrant health center (IHC) affiliated to primary health care institutions.

## 2. Materials and Methods

In order to increase the access of Syrian immigrants to preventive and essential health services in our country, there are immigrant health centers (IHC) affiliated to primary health care institutions where these people live in large numbers. In addition to primary health care services, internal medicine, pediatrics, obstetrics and gynecology, oral and dental health and psychosocial support services are also provided in strengthened IHCs [27]. One of the most important features of these centers is that Syrian healthcare workers are employed in IHCs to eliminate the language barrier for immigrant individuals. Syrian immigrant women living in Turkey receive health care services, including vaccination, for free from the immigrant health center, family health center or hospital during their pregnancy.

Within the scope of the research, vaccination and demographic data of pregnant Syrian women who applied to a strengthened immigrant health center affiliated to the Sultanbeyli District Health Directorate in Istanbul, Turkey, between August 2020 and August 2022 were evaluated. The Sultanbeyli district has the lowest socio-economic development index compared with the other districts of Istanbul and is one of the districts with high numbers of Syrian immigrants in Istanbul [28].

### 2.1. Study Design and Participants

The research is a retrospective-designed cross-sectional type of study. Syrian immigrants can apply to primary health care institutions to receive essential health services in our country as mentioned above. Strengthened immigrant health centers are institutions where essential health care services are provided, such as family health centers. The study sample consists of pregnant Syrian women who received health care from the strengthened immigrant health center in a District Health Directorate in Istanbul, Turkey between August 2020 and August 2022. A calculation of the minimum sample size was not made, while it was aimed to include all of the pregnant Syrian women who applied to the IHC for health services within a time period of approximately 2 years. There were 415 admissions by pregnant Syrian women to the immigrant health center in the time period of 2 years. Inclusion criteria for the study were being a pregnant Syrian woman, being over 18 years old and having an identity number. Seven women who did not meet the inclusion criteria were excluded from the study. Then, the data of the study were evaluated with 408 pregnant Syrian women. 

### 2.2. Measures

The data of pregnant Syrian women who applied to the immigrant health center to receive health services were evaluated within the scope of the study. Demographic data and pregnancy-related health records were examined retrospectively only via system records of the IHC’s clinics without any contact with the pregnant women. Among the evaluated data, there was information about the age of the pregnant woman, trimesters of pregnancy, the number of pregnancies, the status about high-risk pregnancy and the status about vaccination against influenza, COVID-19 and tetanus. Dates of vaccinations and vaccine types of COVID-19 (Biontech and Sinovac) were also evaluated. In the study, full vaccination status for COVID-19 and tetanus was defined as a minimum 2 doses of vaccination separately for both. 

The data evaluated within the scope of the study consisted of health records entered into the IHC system according to the verbal statements and previous health records of pregnant women who applied for health services (such as antenatal care services, vaccination and other primary health care services) at the IHC. In the study, these data were evaluated retrospectively and no scale or questionnaire was used for the evaluation of the data.

### 2.3. Statistical Evaluation and Analysis

SPSS (Statistical Package for Social Sciences) for Windows 25.0 program was used for statistical analysis and data recording. In the study, mean, standard deviation median, minimum and maximum values, numbers (n) and percentages (%) were used for descriptive data. The conformity of continuous variables to normal distribution was examined using visual (histogram and probability charts) and analytical methods (Kolmogorov–Smirnov/Shapiro–Wilk tests). The Mann–Whitney U test was used to compare the two groups in the data that did not fit the normal distribution, and Pearson chi-square test was used to compare the categorical data. The statistical significance level was determined as *p* < 0.05.

## 3. Results

The median age of the 408 pregnant women included in the study was 25.0 (min: 18.0; max: 46.0). The percentages of women in the first, second and third trimesters were 41.9% (n = 171), 47.6% (n = 194) and 10.5% (n = 43), respectively. The median number of pregnancies was 3.0 (min: 1.0; max: 10.0). Of the pregnant women, 8.8% (n = 36) had a high-risk pregnancy in their current pregnancies (Table 1).

The vaccination status of pregnant women was evaluated. None of the pregnant women had received the influenza vaccine. When the status for tetanus vaccination of the pregnant women was evaluated, 149 of the 408 pregnant women were still within the recommended time interval for tetanus vaccination. Of the remaining 259 pregnant women whose tetanus vaccine data were evaluated, 29.7% (n = 77) had received at least two doses of the tetanus vaccine. When the COVID-19 vaccine uptake of the pregnant women was evaluated, 19.4% (n = 79) of the pregnant women were vaccinated with a minimum of two doses of a COVID-19 vaccine. However, 69.3% (n = 283) of the pregnant women had not received the COVID-19 vaccine. Of the women, 11.3% (n = 46) had received only one dose of a COVID-19 vaccine (Table 2). There were no pregnant women who received more than three doses of a COVID-19 vaccine. The brand names of the COVID-19 vaccines for each of the three doses were also evaluated and shown in Figure 1.

Of the pregnant women, 17 (4.2%) had been infected with COVID-19 during their pregnancy. None of these women were fully vaccinated against COVID-19. There were two pregnant women hospitalized due to COVID-19 infection, but none of the hospitalized women were intubated. The history of COVID-19 infection during pregnancy and pregnancy trimesters of the PCR-positive pregnant women are shown in Table 3.

When the factors that may be associated with COVID-19 vaccine uptake were examined, no significant relationship was found between maternal age and those with high-risk pregnancies and COVID-19 vaccine uptake (*p* = 0.097; *p* = 0.634). However, pregnant women with COVID-19 vaccine uptake had significantly higher numbers of pregnancies (*p* = 0.007). Additionally, the percentage of COVID-19 vaccination in the first trimester was significantly higher than in the second and third trimesters (*p* = 0.013) (Table 4).

## 4. Discussion

The number of Syrian immigrants in our country is increasing and it is extremely important to evaluate the frequency and quality of health services received by this population. In this way, appropriate interventions will be made in the areas that immigrants need and the wellbeing of immigrants will be increased. Immunization is one of the essential health care services. In the current study, the aim was to evaluate the vaccine uptake of pregnant Syrian women who received health care services from the strengthened Immigrant Health Center (IHC) during the COVID-19 pandemic.

The vaccination of pregnant women is one of the main strategies recommended by WHO to eliminate tetanus disease [29]. A tetanus vaccine is necessary during pregnancy to prevent both maternal and neonatal tetanus [24]. Research indicates that the immunization of pregnant women or women of childbearing age with at least two doses of the tetanus vaccine is estimated to reduce mortality from neonatal tetanus by 94% [30]. In our study of pregnant women, 29.7% had received at least two doses of the tetanus vaccine. According to the Turkey Demographic and Health Survey (TNSA) 2018, 81.0% of Turkish pregnant women were vaccinated against tetanus for their last live birth in the last 5 years [24]. Only 30.0% of pregnant Syrian women were vaccinated against tetanus for their last birth [25]. There is a need to increase tetanus vaccination coverage in pregnant Syrian women.

According to the data of the Turkish Ministry of Health, 85.7% of the individuals aged 18 years and older received at least two doses of a COVID-19 vaccine [31]. In a study conducted with immigrants in our country, in which almost all of the participants were Syrian immigrants, 68.2% of the participants received at least two doses of a COVID-19 vaccine [32]. Being infected with COVID-19 during pregnancy increases the risk of complications in both the pregnant woman and baby [8]. For this reason, COVID-19 vaccines are also recommended to pregnant women [9]. The vaccination of perinatal women against COVID-19 is seen as an important weapon against COVID-19 infection [33]. However, vaccine hesitancy is a huge barrier to vaccine uptake [34]. In a community-based study in Turkey, 45.3% of the participants were hesitant about receiving the COVID-19 vaccine [35]. In a recent study, the intentions for COVID-19 vaccination among immigrants were found to be lower than the local population [36]. In the same study, nearly half of the participants thought that the COVID-19 vaccine could not be received by pregnant women. In another study, ethnic minority pregnant women were twice as likely to reject a COVID-19 vaccine [37]. It is stated that COVID-19 vaccine coverage in pregnant women is lower than in non-pregnant women [38]. The percentage of pregnant women with two doses of a COVID-19 vaccine was 19.4% in our study. According to data from the CDC, approximately 31% of pregnant women have received the COVID-19 vaccine [4]. Our results indicate that immigrants may also have a lower vaccine uptake against COVID-19 when compared with the data of the CDC. It is necessary to increase the COVID-19 vaccination uptake in pregnant women and especially in immigrants from vulnerable groups. Belief in the benefit of the vaccine affects vaccine uptake in pregnant women [39]. In a study conducted in Turkey, pregnant women who received recommendations for vaccination by health personnel had lower vaccine hesitancy [40]. Evidence-based information about vaccination should be given to pregnant women by health professionals, especially in primary health care institutions. In this way, the possible barriers against vaccination caused by false beliefs or lack of knowledge will be resolved.

In a study, immigrants from ten countries were screened for COVID-19 and 15.1% of them were positive for COVID-19 upon testing [41]. In another study conducted in our country, 7.7% of screened pregnant women were positive for COVID-19 [42]. In the current study, 4.2% of the pregnant women had a COVID-19 infection during their pregnancy period. The slightly lower prevalence of COVID 19 infection in pregnant immigrant women in our study may be due to the low number of confirmed cases due to immigrant women’s inability to use health services adequately [43]. According to several studies, host genetic factors are thought to be one of the factors that play a role in susceptibility to become infected with COVID-19 [44,45]. Genetic factors may also have played a role in the different results in terms of COVID-19 infection positivity among the studies. Participants from different races may also have different genetic characteristics that predispose them to COVID-19 infection. On the other hand, none of the 17 pregnant women with a COVID-19 infection were fully vaccinated in our study. According to a study conducted in Scotland, 77.4% of COVID-19 infections in pregnancy occurred in unvaccinated women, 11.5% in partially vaccinated women and 11.1% in fully vaccinated women [46]. This result draws attention to the fact that the vaccine is effective in preventing COVID-19 infection, similarly to our results. Most of the COVID-19-positive pregnant women in our study were in the last trimester. This may be due to screening for COVID-19 infection prior to interventional procedures, such as cesarean delivery, in our country. 

Seasonal influenza is one of the most common infections and pregnant women are at high risk of morbidity and mortality [47]. Despite the availability of a vaccine against influenza, the rate of vaccination in pregnant women is very low. According to a systematic review, the percentages of seasonal influenza vaccination among pregnant women ranges from 1.7 to 88.4% [47]. To our knowledge, there is no study evaluating the percentages of influenza vaccination in pregnant Syrian women. According to the results of a study conducted in our country, 2.2% of pregnant women had an influenza vaccination during pregnancy [48]. In another study in our country, 8.3% of pregnant women received an influenza vaccine during their pregnancies [49]. According to a study conducted in Italy, the influenza vaccination rates of Italian and pregnant immigrant women was 6.7% and 4.5%, respectively. According to all the information mentioned above, vaccine uptake against influenza is very low both in pregnant immigrant and non-immigrant women. In current study, there were no pregnant Syrian women vaccinated against influenza. Although influenza vaccination percentages in pregnant women are low in the literature, it is worrying that none of the pregnant Syrian women in our study were vaccinated against influenza. According to the results of the studies in the literature, pregnant women unvaccinated against influenza stated mostly the fear of harm to their babies, the lack of knowledge about the vaccine and the thought that the vaccine is unnecessary as reasons for not getting vaccinated [48,49]. In further studies to be carried out, the reasons for the low influenza vaccination percentages of pregnant Syrian women should be investigated and intervention strategies should be developed for them. Qualitative studies can be conducted with pregnant women to understand why there is no influenza vaccine uptake in pregnant Syrian women.

In our study, some parameters that can affect vaccine uptake against COVID-19 were examined. According to our results, pregnant women with COVID-19 vaccine uptake had a significantly higher number of pregnancies. Similarly, vaccination coverage for influenza was higher among women who had been pregnant before [39]. Additionally, the uptake of a COVID-19 vaccination in the first trimester was significantly higher than in the second and third trimesters in our study. According to a recent study, 85.7% of the pregnant women vaccinated with COVID-19 had received the vaccine in the third trimester and 14.3% in the second trimester of pregnancy [38]. Data are needed to make clear why different results are seen about COVID-19 vaccine uptake times among pregnant women.

Some parameters that may affect the vaccination status of pregnant women, such as educational status, income or presence of chronic diseases, could not be evaluated in the study. This is one of the limitations of our study. There is a need for further qualitative and quantitative studies in which the barriers against the use of vaccination services by pregnant immigrant women are examined. Another limitation is that the study was not conducted as population-based. It was carried out on the vaccination data of pregnant women who applied to a primary health care institution. However, about one-third of Syrian women do not use health services according to the literature [50]. The percentages of vaccine uptake may probably be lower in pregnant women who do not apply to a health institution. There are many studies in the literature that evaluate access and utilization of health services by immigrants. However, in the literature, the number of studies evaluating the data of pregnant immigrant women, influenza, tetanus and COVID-19 vaccinations, and history of COVID-19 infection is very limited. In our study, we were able to evaluate all these data together and provide a broad perspective on this topic. As we know, there is no study evaluating the percentage of influenza vaccinations in pregnant Syrian women, so we think that we have brought novelty to the literature in this field. The evaluation of these data in our study is the strength of this study.

## 5. Conclusions

Immigrants are one of the most vulnerable and disadvantaged populations in terms of the utilization of health services. The results of current study show that vaccine uptake in pregnant immigrant women is very low. Language problems, insufficient economic conditions and low educational status can be potential causes for that [51]. Increasing the utilization of vaccination services by pregnant immigrant women is an extremely important issue for the health of pregnant women and their babies. More innovative strategies and public health interventions are needed to improve vaccination coverage among disadvantaged groups [52]. For this reason, we suggest that future studies may be more focused on understanding the barriers to vaccine uptake and improving strategies promoting vaccination and reducing the inequalities in the utilization of healthcare services among pregnant immigrant women.

## Figures and Tables

**Figure 1 vaccines-11-00257-f001:**
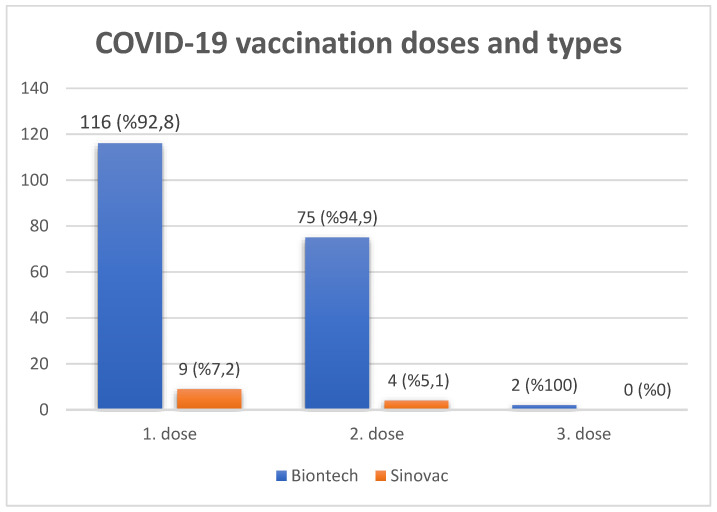
COVID-19 vaccination doses and types. No women were vaccinated with Turkovac.

**Table 1 vaccines-11-00257-t001:** Age and pregnancy-related features of pregnant women (n = 408).

Features	
Age, median (min-max)	25.0 (18.0–46.0)
Trimester, n (%)	
First	171 (41.9)
Second	194 (47.6)
Third	43 (10.5)
Number of pregnancies, median (min-max)	3.0 (1–10)
High-risk pregnancy, n (%)	36 (8.8)

**Table 2 vaccines-11-00257-t002:** Vaccine uptake and frequency of COVID-19 infection of pregnant women.

Influenza vaccination	n (%)
Yes	0 (0)
No	408 (100.0)
Tetanus vaccination (at least 2 doses)	n (%)
Yes	77 (29.7)
No	182 (70.3)
COVID-19 vaccination (at least 2 doses)	n (%)
Yes	79 (19.4)
Two doses	77 (18.9)
Three doses	2 (0.5)
No	329 (80.6)
No vaccine	283 (69.3)
Single dose	46 (11.3)

**Table 3 vaccines-11-00257-t003:** History of COVID-19 infection during pregnancy and pregnancy trimesters of PCR-positive pregnant women.

History of COVID-19 infection	n (%)
Yes	17 (4.2)
No	391 (95.8)
COVID-19 PCR-positive pregnant women	n (%)
Trimester	
First	2 (11.8)
Second	4 (23.5)
Third	11 (64.7)

**Table 4 vaccines-11-00257-t004:** COVID-19 vaccine uptake of pregnant women.

	**COVID-19 Vaccine Uptake**		
	No	Yes	*p* Value
	Median (Min-Max)	Median (Min-Max)	
	Mean ± SD	Mean ± SD	
Age	25.0 (18.0–46.0)	27.0 (18.0–41.0)	0.097 *
	26.08 ± 5.96	27.09 ± 5.60	
Number of pregnancies	3.0 (1.0–9.0)	3.0 (1.0–10.0)	0.007 *
	2.99 ± 1.55	3.73 ± 2.10	
	n (%)	n (%)	*p* value
Trimester			0.013 **
First	127 (74.3)	44 (25.7)	
Second	163 (84.0)	31 (16.0)	
Third	39 (90.7)	4 (9.3)	
High-risk pregnancy			0.634 **
Yes	8 (77.8)	8 (22.2)	
No	274 (81.1)	64 (18.9)	

* Mann–Whitney U test, ** Pearson Chi-Square Test.

## Data Availability

Not applicable.

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
