# Peer review of "Vaccine Uptake and COVID-19 Frequency in Pregnant Syrian Immigrant Women"

_vaccines, 2023, doi:10.3390/vaccines11020257_

Round 1
Reviewer 1 Report
This is a helpful paper raising an important issue.
I realise the authors may be working in their second or third language but the English needs quite a bit of work to make it easy to read and understand
for example"Pregnants" in the title and elsewhere should be changed to "women who are pregnant" or "pregnant women".
"Vaccine uptake among immigrant pregnant women is insufficient" It is better to say "very low" as you don’t define what is sufficient
Line 68 "the rate of pregnant women who are not vaccinated against tetanus is more than 4 times that of vaccinated pregnants" I am not sure what this means. Do you mean the percentage? It would also be helpful to give the percentage.
Line 45 does this refer to Covid vaccination or all vaccinations
line 217 and around there. There is no mention of the % of influenza immunisation in Turkish pregnant women to compare with the Syrians
The abstract needs improved. It does not need to mention SPSS (Statistical Package for Social Sciences) for Windows 25.0 program was used for statistical 17 analysis.
Author Response
Point 1: I realise the authors may be working in their second or third language but the English needs quite a bit of work to make it easy to read and understand for example"Pregnants" in the title and elsewhere should be changed to "women who are pregnant" or "pregnant women". "Vaccine uptake among immigrant pregnant women is insufficient" It is better to say "very low" as you don’t define what is sufficient.
Response 1: Thank you very much for your valuable comment. We read the manuscript more carefully and done some changes for the words as you suggested. You can see the corrections in the main manuscript that we uploaded.
Point 2: Line 68 "the rate of pregnant women who are not vaccinated against tetanus is more than 4 times that of vaccinated pregnants" I am not sure what this means. Do you mean the percentage? It would also be helpful to give the percentage.
Response 2: Thank you very much, we mean the percentage in this sentence. We have changed the word as "percentage" in the main manuscript. You can see this correction in the revised manuscript that we uploaded.
Point 3: Line 45 does this refer to Covid vaccination or all vaccinations
Response 3: Thank you, it refer to only Covid vaccination.
Point 4: line 217 and around there. There is no mention of the % of influenza immunisation in Turkish pregnant women to compare with the Syrians
Response 4: Thank you for your comment. We added some information about the percentage of influenza immunisation in Turkish pregnant women. You can see in the revised manuscript.
Point 5: The abstract needs improved. It does not need to mention SPSS (Statistical Package for Social Sciences) for Windows 25.0 program was used for statistical 17 analysis.
Response 5: Thank you for your comment. We have done some improvements in the abstract part as you have suggested. In the revised manuscript; we did not give the information about the program for statistical analysis in the abstract part. You can see the changes in the revised manuscript that we uploaded.
Reviewer 2 Report
1.It is suggested to add new research studies about the side effects of vaccines.
2.what is the suggestion of this study for future works?
3.Please discuss about the type of vaccines and properties.
4.It will be better to add the role of genetic factors.
5.More references for the discussion part of manuscript and bold your study novelty should be added: e.g.,
-DOI: 10.3390/vaccines10091533
-DOI:10.3390/microorganisms9020232
-DOI:10.54005/geneltip.1147493
Author Response
Point 1: It is suggested to add new research studies about the side effects of vaccines.
Response 1: Thank you! We added studies about the side effects of vaccines in the Introduction part. You can see in the main manuscript.
Point 2: what is the suggestion of this study for future works?
Response 2: Thank you for your comment. In the conclusion part, we mentioned our suggestion for future works. You can see in the main manuscript.
Point 3: Please discuss about the type of vaccines and properties.
Response 3: Thank you so much for your suggestion. We added information about the the type of vaccines in the Introduction part, as you have suggested. You can see the changes in the main manuscript.
Point 4: It will be better to add the role of genetic factors.
Response 4: Thank you for your sugegstion. We added information about the genetic factors and COVID-19 infection. You can see in the fourth paragraph of discussion part.
Point 5: More references for the discussion part of manuscript and bold your study novelty should be added: e.g.,
-DOI: 10.3390/vaccines10091533
-DOI:10.3390/microorganisms9020232
-DOI:10.54005/geneltip.1147493
Response 5: Thank you for your suggestions. We added more references for the discussion part as you suggested. And the novelty of our study was highlighted at the end of the discussion part just before the conclusion part.
Reviewer 3 Report
Suggestions:
1. Improve the overall writing style of the article
2. Methodology: Provide details about the Data collection tool, How it was designed? Schedule or Questionnaire? Validation? Method of sampling
3. Do not start sentences with numbers.
4. Add more studies in discussion and try to cite reasons for similar or contrasting findings
Author Response
Point 1: Improve the overall writing style of the article
Response 1: Thank you so much. We have done some improvements in the overall writing style of the article. You can see in the main manuscript.
Point 2: Methodology: Provide details about the Data collection tool, How it was designed? Schedule or Questionnaire? Validation? Method of sampling
Response 2: Thank you so much. More details about data collection and sampling were provided. We hope it is more clear now. You can see changes in the "Study Design and Participants" and "Measures" sections of Materials and Methods part of the main manuscript.
Point 3: Do not start sentences with numbers.
Response 3: Thank you very much. As you mentioned, we have corrected all sentences starting with numbers. You can see in the main manuscript.
Point 4: Add more studies in discussion and try to cite reasons for similar or contrasting findings
Response 4: Thank you for your suggestion. More studies were added in the discussion part and reasons were cited for similar or contrasting findings. You can find the changes in the main manuscript.